# *Panax notoginseng* Suppresses Bone Morphogenetic Protein-2 Expression in EA.hy926 Endothelial Cells by Inhibiting the Noncanonical NF-κB and Wnt/β-Catenin Signaling Pathways

**DOI:** 10.3390/plants11233265

**Published:** 2022-11-28

**Authors:** Tsu-Ni Ping, Shu-Ling Hsieh, Jyh-Jye Wang, Jin-Bor Chen, Chih-Chung Wu

**Affiliations:** 1Department of Food and Nutrition, Providence University, Taichung 43301, Taiwan; 2Department of Seafood Science, National Kaohsiung University of Science and Technology, Kaohsiung 81157, Taiwan; 3Department of Nutrition and Health Science, Fooyin University, Kaohsiung 83102, Taiwan; 4Division of Nephrology, Department of Internal Medicine, Kaohsiung Chang Gung Memorial Hospital and Chang Gung University College of Medicine, Kaohsiung 83301, Taiwan

**Keywords:** *Panax notoginseng* water extracts, bone morphogenetic protein-2, vascular calcification

## Abstract

*Panax notoginseng* (PN) exerts cardiovascular-disease-protective effects, but the effect of PN on reducing vascular calcification (VC) is unknown. Under the VC process, however, endothelial bone morphogenetic protein-2 (BMP-2) signals connect endothelial and smooth muscle cells. To investigate the effects of PN water extract (PNWE) on BMP-2 expression, human EA.hy926 endothelial cells were pretreated with PNWE for 48 h, and BMP-2 expression was then induced using warfarin/β-glycerophosphate (W/BGP) for another 24 h. The expression of BMP-2, the degrees of oxidative stress and inflammation, and the activation of noncanonical NF-κB and Wnt/β-catenin signaling were analyzed. The results showed that the BMP-2 levels in EA.hy926 cells were reduced in the groups treated with 10, 50, or 100 μg/mL PNWE combined with W/BGP. PNWE combined with W/BGP significantly reduced thiobarbituric-acid-reactive substrate and reactive oxygen species levels as well as prostaglandin E2, IL-1β, IL-6, and TNF-α. PNWE (10, 50, and 100 μg/mL) reduced the p52 levels and p52/p100 protein ratio. Wnt and β-catenin protein expression was decreased in the groups treated with PNWE combined with W/BGP. These results showed that PNWE reduced BMP-2 expression in EA.hy926 cells by inhibiting the noncanonical NF-κB and Wnt/β-catenin signaling pathways.

## 1. Introduction

*Panax notoginseng* (PN) is not only a Chinese medicine that is used to treat cardiovascular disease (CVD) but is also a dietary supplement. Previous studies have shown that PN exerts many physiological effects; PN is used to treat atherosclerosis [1], regulate blood sugar [2], inhibit cancer growth [3] and metastasis [4,5], protect kidneys [6], and treat depression [7,8]. Notably, previous studies on the role of PN in alleviating cardiovascular pathology showed that PN can reduce vascular endothelial disorders in C57BL/6 mice fed a high-fat diet by decreasing the phosphorylation of 5’AMP-activated protein kinase (AMPK), the expression of endothelial nitric oxide synthase (eNOS), endoplasmic reticulum stress, and oxidative stress [9]. Saponins from PN can also suppress the progression of atherosclerotic lesions by inhibiting the NF-κB signaling pathway and inhibiting IL-6, IL-1β, TNF-α, and Calpain1 protein expression in the aortic root tissues of apoE−/− mice [1]. Among the components of PN, ginsenoside Rb1 and Rg1 spontaneously regulate blood pressure in hypertensive rats through the activation of nitric oxide (NO) by modulating the PI3K/Akt/eNOS pathway and l-arginine transport in endothelial cells [10]. Except for some reducing atherosclerosis studies, the effects of PN on arteriosclerosis and vascular calcification (VC) are still unclear.

In clinical diagnosis, VC has typically been used as an indicator to assess cardiovascular and renal disease [7]. However, VC is also a progressive disease, and it is usually a comorbidity of chronic kidney disease (CKD), CVD, and diabetes mellitus (DM); VC leads to poor prognosis [11] and high mortality rates in CKD and CVD patients [12]. Approximately 40% of patients with end-stage renal disease (ESRD) die due to CVD. ESRD patients have complications of CVD, and these complications usually have a strong association with VC [13]. VC is one manifestation of arteriosclerosis but not atherosclerosis. VC represents a pathological change in vascular smooth muscle tissues, not the accumulation of lipids and cholesterol [14]. VC involves fibrosis and thickening of the medial arterial layer during vascular pathophysiological processes [15]. VC progression involves signals connecting the endothelium and smooth muscle cells in vascular tissue [15]. When the vascular endothelium experiences disordered calcium and phosphorus homeostasis, oxidative stress and chronic inflammation, bone morphogenetic protein-2 (BMP-2) is expressed by endothelial cells and released to smooth muscle cells [16]. BMP-2 plays an important role in signal transduction and communication between endothelial cells and smooth muscle cells [16].

In VC progression, endothelial BMP-2 signaling is required for runt-related transcription factor 2 (RUNX2)-dependent induction of osteoblast-like smooth muscle cells [17]. BMPs are potent osteogenic agents that stimulate the maturation of mesenchymal osteoprogenitor cells to osteoblasts [18]. This stimulation involves two types of signal transduction pathways, including canonical drosophila mothers against the decapentaplegic (SMAD) pathway [19] and mitogen-activated protein kinase (MAPK) pathways [20], leading to the transformation of vascular smooth muscle cells into osteoblast-like cells [18]. Therefore, reducing BMP-2 expression and activity by reducing oxidative stress and providing anti-inflammatory effects may be one method of alleviating VC. BMP-2 expression is transcriptionally regulated by the Wnt/β-catenin [21] and noncanonical nuclear factor kappa-light-chain-enhancer of activated B (NF-κB) [22] signaling pathways. Moreover, oxidative stress and inflammation can regulate the activation of canonical and noncanonical NF-κB signaling [23,24] and Wnt/β-catenin signaling [25].

The aim of this study was to investigate the regulatory effect of PN on endothelial cell BMP-2 expression. To study the potential of PN in reducing VC progression, we prepared PN water extract (PNWE) through ultrasonic extraction, vacuum concentration, and lyophilization as an experimental material in this study. Warfarin/beta-glycerophosphate (W/BPG) was used as the induction model. To determine the effect of PNWE on BMP-2 expression and regulation, BMP-2 protein levels, oxidative stress, and the inflammatory response were analyzed. To investigate the regulation of BMP-2 transcription by PNWE, the activation of Wnt/β-catenin and noncanonical NF-κB signaling was measured.

## 2. Results

### 2.1. PNWE Protects EA.hy926 Cells from W/BGP-Induced Damage

The viability of EA.hy926 cells treated with 10, 50, or 100 μg/mL PNWE for 24, 48, or 72 h was not significantly different from that of control cells (Figure 1A). When EA.hy926 cells were treated with W/BPG for 72 h, the cell viability was 79.6 ± 2.7%, which was significantly lower than that of control cells (100%) (*p* < 0.05), and the cell viability of the 100 μg/mL PNWE group was not different from that of the control group (Figure 1B). This result means that W/BGP may decrease cell viability, but 100 μg/mL PNWE alone did not cause damage. However, the cell viability of the group treated with 100 μg/mL PNWE combined with W/BGP (91.7 ± 1.0%) was significantly higher than that of the group treated with W/BGP alone (79.6 ± 2.7%) (*p* < 0.05) and was not different from that of the control group (100%). The results showed that 100 μg/mL PNWE improved the viability of EA.hy926 cells after damage caused by W/BGP (Figure 1B). The cell morphological examination showed that the W/BGP-treated group exhibited slight deformation and fracture (Figure 1C). There were no significant differences in the cell number and morphology between the control group and each of the groups treated with any concentration of PNWE combined with W/BGP.

### 2.2. PNWE Decreased W/BGP-Induced BMP-2 Protein Expression in EA.hy926 Cells

Immunocytochemistry and western blotting analyses were used to determine the distribution and expression of BMP-2 in EA.hy926 cells treated with W/BGP (Figure 2). The immunocytochemistry assay showed that when W/BGP was incubated with EA.hy926 cells, the fluorescence intensity of intracellular BMP-2 was higher than that in the control group, the group treated with PNWE combined with W/BGP, and the group treated with PNWE alone (Figure 2A). The quantitative results in Figure 2B show that W/BGP treatment significantly increased BMP-2 protein expression by 47 ± 7.1%, and treatment with 10, 50, and 100 µg/mL PNWE combined with W/BGP dramatically reduced the BMP-2 levels by 22–43% compared to those in the W/BGP-treated group (*p* < 0.05).

The immunoblotting analysis results also showed that the protein levels of BMP-2 in the W/BPG group were significantly higher than those in the control group (*p* < 0.05) (Figure 2C). The PNWE group did not exhibit increased intracellular BMP-2 levels. After treatment with 10, 50, or 100 µg/mL PNWE combined with W/BGP, the intracellular BMP-2 levels were considerably reduced by 18–21% compared to those in the W/BGP-treated group (*p* < 0.05) (Figure 2C,D).

These results showed that PNWE could decrease the W/BPG-induced intracellular BMP-2 protein levels.

### 2.3. PNWE Decreased W/BGP-Induced Oxidative Stress in EA.hy926 Cells

To measure the antioxidative effects of PNWE on EA.hy926 cells treated with W/BGP, the intracellular 2-thiobarbituric acid reacting substances (TBARS) levels and reactive oxygen species (ROS) levels in EA.hy926 cells were analyzed. Figure 3A shows that the TBARS levels of the W/BGP group (0.45 ± 0.2 nmol/mg protein) were significantly higher than those of the control group (0.21 ± 0.1 nmol/mg protein) (*p* < 0.05). However, 100 μg/mL PNWE alone did not change the TBARS levels compared to the control. The TBARS levels in EA.hy926 cells treated with 10, 50, or 100 μg/mL PNWE combined with W/BGP (0.30 ± 0.2, 0.24 ± 0.2 or 0.15 ± 0.2 nmol/mg protein, respectively) were significantly lower than those in the group treated with W/BGP alone (0.45 ± 0.2 nmol/mg protein) (*p* < 0.05).

In addition, the ROS levels in the W/BGP group were significantly increased by 18% compared to those in the control group (*p* < 0.05) (Figure 3B). This result showed that W/BPG could truly lead to oxidative stress. However, when EA.hy926 cells were treated with 10, 50, or 100 μg/mL PNWE combined with W/BGP (approximately 82–84%), the ROS levels (100%) were significantly lower than those in the W/GBP-treated group (*p* < 0.05) (Figure 3B).

These results showed that PNWE could decrease W/BPG-induced oxidative stress in EA.hy926 cells.

### 2.4. PNWE Alleviated the W/BGP-Induced Inflammatory Response in EA.hy926 Cells

The inflammatory response in EA.hy926 cells treated with W/BGP was assessed. The NO content was not affected in the control, W/BGP, PNWE combined with W/BGP, or PNWE groups (*p* < 0.05) (Figure 4A). The *prostaglandin E2* (PGE_2_) content of EA.hy926 cells was significantly increased after W/BGP treatment (100%) compared to the control treatment (62.3 ± 0.8%) (*p* < 0.05). After treatment with 10, 50, or 100 μg/mL PNWE combined with W/BGP (71.0 ± 9.8, 74.9 ± 5.4, and 79.9 ± 10.0%, respectively), the PGE_2_ content was significantly decreased compared to that in the W/BPG-treated group (100%) (*p* < 0.05) (Figure 4B).

Moreover, Figure 4C–E show that the W/BGP-treated group had significantly increased IL-1β, IL-6, and TNF-α contents (increased by 71, 49, and 17%, respectively) compared to the control group (*p* < 0.05). However, the contents of IL-1β, IL-6 and TNF-α in EA.hy926 cells treated with 10, 50 or 100 μg/mL PNWE combined with W/BGP were significantly decreased by 39–48, 34–44 and 17–21%, respectively, compared to those in the W/BGP-treated group (*p* < 0.05). No differences were observed in the NO, PGE_2_, IL-1β, IL-6, and TNF-α contents after EA.hy926 cells were treated with 100 μg/mL PNWE alone compared to the control.

These results showed that PNWE alleviated the inflammatory response in EA.hy926 cells treated with W/BGP. These concentrations of PNWE did not induce an inflammatory response in EA.hy926 cells.

### 2.5. PNWE Decreased the W/BGP-Induced Activation of the Wnt3a/β-Catenin Signaling Pathway in EA.hy926 Cells

In this study, Wnt3a protein expression was significantly increased in EA.hy926 cells treated with W/BGP (*p* < 0.05) (Figure 5A,B). However, when EA.hy926 cells were treated with 10, 50, or 100 µg/mL PNWE combined with W/BGP, Wnt3a levels were significantly decreased by 24–37% compared to those in the W/GBP-treated group (*p* < 0.05) (Figure 5A,B). In addition, W/BGP also significantly increased β-catenin expression, and PNWE decreased the W/BGP-induced intracellular β-catenin levels (Figure 5C,D). When EA.hy926 cells were treated with 50 or 100 µg/mL PNWE combined with W/BGP, they exhibited significantly reduced β-catenin levels (decreased by 16 and 39%, respectively) compared to the W/GBP-treated group (*p* < 0.05) (Figure 5C,D).

### 2.6. PNWE Decreases the W/BGP-Induced Activation of the Noncanonical NF-κB Signaling Pathway in EA.hy926 Cells

Figure 6A,B show that the p52 content significantly increased after W/BGP treatment compared to the control (*p* < 0.05). Conversely, when EA.hy926 cells were treated with 0, 50 or 100 µg/mL PNWE combined with W/BGP, the p52 protein content was significantly decreased compared to that in the W/BPG-treated group (*p* < 0.05) (Figure 6A,B).

Furthermore, the intracellular p-p100 expression assay results showed that W/GBP also significantly increased the p-p-100 levels compared to the control (*p* < 0.05) (Figure 6C,D), and 50, and 100 µg/mL PNWE could decrease the W/BGP-induced p-p-100 levels (*p* < 0.05) (Figure 6C,D). In addition, the p52/p100 protein ratio was significantly increased after W/BPG induction compared to the control, and 10, 50, and 100 µg/mL significantly decreased the p52/p100 protein ratio compared to the W/BPG group (*p* < 0.05) (Figure 6E).

These results showed that PNWE could inhibit noncanonical NF-κB signaling activation by reducing Wnt and β-catenin expression.

## 3. Discussion

This study demonstrated that PNWE suppressed intracellular BMP-2 expression in W/BGP-treated EA.hy926 cells. As Figure 7 shows, PNWE decreased the W/BGP-induced oxidative stress and inflammatory response and inhibited NF-κB and Wnt/β-catenin signaling, leading to the suppression of BMP-2 transcription and expression. This result means that PNWE has the potential to alleviate the risk of VC by reducing BMP-2 expression.

CKD, DM, and CVD patients usually have abnormal calcium and phosphorus homeostasis, oxidative stress, or chronic inflammation [16]. These disorders promote BMP-2 expression in endothelial cells and its release to smooth muscle cells, which leads to the transformation of smooth muscle cells into osteoblast-like cells and results in VC [26]. Therefore, regulating BMP-2 expression by active dietary components in addition to clinical treatment may be a solution for VC prevention and improvement. Several studies have shown that suppressing intracellular BMP-2 expression via antioxidant approaches and inhibiting the inflammatory response could limit the activation of VC. For example, baicalein, which is a natural flavonoid extract of *Scutellaria baicalensis* rhizome, can reduce BMP-2 expression by decreasing the levels of malondialdehyde, which is a product of lipid peroxidation, and increasing the activity of superoxide dismutase in Sprague Dawley rats [27]. In a human umbilical vein endothelial cell culture model, gemigliptin inhibited BMP-2 by reducing IL-1β levels and suppressing the endothelial-to-mesenchymal transition during VC progression [28]. Phloretin, which is a plant-derived dihydrochalcone compound, effectively reduces MCP-1 and BMP-2 levels in human umbilical vein endothelial cells (HUVECs) stimulated with high glucose. This may be a potential therapeutic approach for treating diabetic endothelial damage and subsequent cardiovascular complications [29]. In addition, *Panax ginseng*, a Chinese medicine, can also reduce vascular calcification by reducing oxidative stress and inflammation. Its steroid glycosides and triterpene saponins are involved in antivascular calcification [30]. A previous study showed that PN saponins from PN can suppress atherosclerosis, reduce NF-κB signaling activation and inhibit proinflammatory factors, including IL-6, IL-1β, TNF-α, and Calpain1 protein expression [1]. Notoginsenoside R1, a specific monoterpene from PN, can reduce inflammation in EA.hy926 cells under oxLDL-induced expression [31]. Li et al. [32] also showed that notoginsenoside R1 can reduce inflammation by reducing TNF-α and IL-6 NF-κB signaling. In this study, we found that PNWE can reduce BMP-2 expression by decreasing oxidative stress, including reducing TBARS and ROS levels, and decreasing the inflammatory response, including reducing IL-1β, TNF-α, IL-6 and PGE2 levels. These results showed that PNWE could reduce BMP-2 expression by improving oxidative stress and inflammation in EAhy926 cells treated with W/BGP. These saponins and terpenes may be involved in the PNWE-mediated reduction of vascular calcification.

NF-κB is a redox-sensitive transcription factor. The activation of NF-κB signaling is regulated by the intracellular response to oxidative stress and inflammation [33]. NF-κB signaling transduction plays an important role in BMP-2 [34], IL-1β [35], IL-6 [36], and TNF-α [37] gene transcription. Both the canonical (NF-κB, p65/p50) and noncanonical NF-κB (p100/p52) signaling pathways are involved in the activation of the BMP-2 gene promoter region and gene transcription [38]. Our previous study showed that PNWE reduced NF-κB (p65/p50) signaling activation by increasing the phosphorylation of I-κB, reducing NF-κB translocation to the nucleus and reducing NF-κB DNA binding activity in EA.hy926 cells [4]. In the present study, PNWE inhibited the noncanonical NF-κB (p100/p52) signaling pathway by reducing intracellular p52 levels, the phosphorylation of p100, and the p52/p100 protein expression ratio in EA.hy926 cells. These results showed that PNWE could inhibit both the canonical and noncanonical NF-κB signaling pathways, not only reducing IL-1β, IL-6, and TNF-α transcription and translation but also reducing BMP-2 transcription and translation. Previous studies also showed that inhibiting both the canonical (NF-κB, p65/p50) and noncanonical NF-κB (p100/p52) signaling pathways could inhibit BMP-2 transcription and expression. For example, in primary coronary arterial endothelial cells from male Wistar rats and a human umbilical vein endothelial cell culture model, vascular BMP-2 expression could be regulated by the H_2_O_2_-mediated activation of both the canonical and noncanonical NF-κB pathways [39]. When TNF-α was used to treat human coronary arterial endothelial cells (HCAECs), resveratrol reduced the expression of NF-κB (p65/p50)-dependent inflammatory markers, such as inducible nitric oxide synthase, IL-6, BMP-2, intercellular adhesion molecule 1 (ICAM-1), and vascular cell adhesion protein (VCAM), via its cardioprotective effects [40]. Zhang et al. [41] showed that pioglitazone, which is a clinical drug used in the treatment of DM, suppressed BMP-2 expression and NF-κB (p65/p50) activation in HUVECs exposed to high glucose. Experiments in a mouse model of alymphoplasia (aly/aly), which is characterized by abnormal p100 and p52 gene expression and metabolism, showed that alternative signaling pathways to the NF-κB pathway via the processing of p52 from p100 could negatively regulate osteoblastic differentiation and bone formation by modifying BMP activity [22]. The present study and previous studies have shown that reduced NF-κB signaling activation regulates BMP-2 expression and that VC plays an important role.

In addition to the NF-kB signaling pathway, BMP expression is regulated by the Wnt/β-catenin signaling pathway [21]. The Wnt signaling pathway is essential for cell proliferation, differentiation, migration, survival, and other processes [21]. Rong et al. [42] reported that BMP-2 plays a crucial role in calcium deposition in vascular smooth muscle cells (VSMCs) and in VC in CKD patients via a mechanism that involves the Wnt/β-catenin pathway. Recently, pioglitazone, which is a drug that is used for blood sugar regulation in DM patients, was shown to alleviate the β-glycerophosphate-induced calcification of rat aortic VSMCs by inhibiting the activity of the Wnt/β-catenin signaling pathway [43]. The present study found that PNWE can decrease W/BGP-induced Wnt and β-catenin expression in EA.hy926 cells. Furthermore, PNWE reduced BMP-2 expression by inhibiting Wnt/β-catenin signaling. Moreover, PNWE significantly decreased the W/BGP-induced TNF-α levels in EA.hy926 cells. TNF-α can inhibit inflammation and regulate the NF-κB and Wnt/β-catenin signaling pathways because the NF-κB and Wnt/β-catenin signaling pathways are activated during TNF-α-induced inflammatory responses [44]. These results demonstrated that PNWE inactivated the NF-κB and Wnt/β-catenin pathways, causing reduced expression of BMP-2 and inflammatory cytokines, including TNF-α, IL-1β, and IL-6, and these factors cross-regulated each other to modulate oxidative stress, inflammation, and BMP-2 expression.

In conclusion, PNWE reduced BMP-2 expression in EA.hy926 endothelial cells by inhibiting oxidative stress and inflammation and inhibiting the activation of the noncanonical NF-κB and Wnt/β-catenin signaling pathways. PNWE has a high potential to alleviate the risk of VC by alleviating BMP-2 expression.

## 4. Materials and Methods

### 4.1. Preparation of PN Water Extract (PNWE)

Roots of *P. notoginseng* were purchased from the E-Lin Chinese Medicine Clinic Center (Taichung City, Taichung). The method for PNWE preparation was modified from that described by Hsieh et al. [4]. First, the PN was ground with a high-speed blender. The powder was extracted by ultrasonication (40 K Herz, Deltadelta Ultrasonic Co., Ltd., New Taipei City, Taiwan) with RO water (*v*/*v*: 1/5) for 1 h and allowed to stand for 30 min in a cold 4 °C room. These ultrasonication and incubation steps were repeated 3 times. The extracts were centrifuged at 250× *g* for 10 min at 4 °C. The supernatants were concentrated under a rotary evaporator (water bath at 55 °C) (Eyela, Tokyo Rikakikai Co., Tokyo, Japan). Finally, the concentrated product was dried in a freeze-dryer (Labconco Co. Kansas City, MO, USA) at −43 °C. The percent yield of PNEE was 9.23% (*w*/*w*). The contents of notoginsenoside R1 and ginsenoside Rg1, both indicator components of PNWE prepared, are 13.98 ± 2.60 mg/g PNWE and 34.67 ± 5.51 mg/g PNWE, respectively.

### 4.2. Cell Culture and Treatment

EA.hy926 cells, which are somatic hybrid human vascular endothelial cells, were purchased from the Bioresource Collection and Research Center (Hsinchu, Taiwan). EA.hy926 cells were cultured in RPMI-1640 medium (Gibco, Thermo Fisher Scientific, Inc., Waltham, MA, USA) supplemented with 100 units/mL penicillin, 100 μg/mL streptomycin, and 10% (*v*/*v*) heat-inactivated fetal bovine serum (FBS; Gibco, Thermo Fisher Scientific, Inc.) at 37 °C in a 5% CO_2_ humidified atmosphere.

After a series of preliminary experiments to determine the optimal concentration of PNWE for treating EA.hy926 cells, EA.hy926 cells (1.0 × 10^5^ cells/30 mm culture dish) were incubated for 24 h. Then, the EA.hy926 cells were pretreated with 10, 50, or 100 μg/mL PNWE for 48 h. Next, the method described by Beazley et al. [45] was used to induce EA.hy926 cell calcification by treatment with 10 μM warfarin/1.6 mM β-glycerophosphate (W/BGP; Sigma-Aldrich Co., St. Louis, MO, USA) for another 24 h prior to various biochemical analyses. PNWE and W/BPG were diluted in sterilized RO H_2_O. Cells treated with sterilized RO H_2_O alone for 72 h served as the control group, and cells treated with sterilized RO H_2_O alone for 48 h followed by W/BGP for the last 24 h served as the W/BGP-treated control group. In addition, to understand the effect of PNWE alone on various biochemical analyses, cells treated with 100 μg/mL PNWE for 72 h served as another control group.

### 4.3. Cell Viability Analysis

In this study, the 3-(4,5-dimethyl-2-yl)-2,5-diphenyltetrazolium bromide (MTT) reduction assay was used to measure cell viability. The MTT assay method was performed as described by Denizot and Lang [46]. At the end of the experimental treatment, the reaction was stopped by removing the medium and washing the cells with cold phosphate-buffered saline (PBS; 3.2 mM Na_2_HPO_4_, 0.5 mM KH_2_PO_4_, 1.3 mM KCl, 135 mM NaCl, pH 7.4). Then, the cells were incubated in DMEM supplemented with 0.5 mg/mL MTT (Sigma-Aldrich Co.) for an additional 3 h. Finally, the medium was removed and extracted with isopropanol (Sigma-Aldrich Co.) for 15 min. The optical density (OD) of the isopropanol fraction was measured at a wavelength of 570 nm in a Microplate Biokinetics Reader (Bio-Tek Instruments, Winooski, VT, USA).

A phase-contrast inverted fluorescence microscope was used to evaluate morphological changes in the cells (Olympus IX51, Olympus, Tokyo, Japan).

### 4.4. Intracellular Lipid Peroxidation and Reactive Oxygen Species (ROS) Level Analysis

The intracellular levels of lipid peroxides were determined based on the levels of thiobarbituric acid reactive substance (TBARS) production [47]. At the end of the experimental treatment, the reaction was stopped by removing the medium and washing the cells with cold PBS. Next, the cells were removed with a cell scraper in 100 μL of 20 mM phosphate buffer supplemented with 0.5% Triton X-100 and 10 mM butylated hydroxytoluene (BHT, in ethanol; Sigma-Aldrich Co.). Then, the cells were centrifuged at 10,000× *g* at 4 °C. A volume of 100 μL of the upper cell suspension was collected, and 2 mL of 0.1 N HCl, 0.3 mL of 10% phosphotungstic acid, and 1 mL of 0.7% 2-thiobarbituric acid were added to the suspension. The resulting mixture was heated for 30 min in boiling water, and TBARS was extracted into 5 mL of n-butanol. After centrifugation at 3500× *g* for 15 min at 4 °C, the fluorescence of the butanol layer was measured with a fluorescence microplate reader (Bio-Tek Instruments). The excitation and emission wavelengths were 515 nm and 555 nm, respectively. The total protein concentrations of the EA.hy926 cells were measured as described by Lowry et al. [47].

At the end of the experimental treatment, the reaction was stopped by removing the medium. The medium removed from each group was collected and used to measure the ROS levels. The levels of ROS in EA.hy926 cells were examined using a Cellular ROS Assay Kit (ab113851, Abcam Inc., Cambridge, MA, USA) according to the manufacturer’s instructions.

### 4.5. Intracellular NO and Prostaglandin E_2_ (PGE_2_) Determination

At the end of the experimental treatment, the reaction was stopped by removing the medium. The medium removed from each group was used to measure the NO and PGE_2_ levels. In this study, the nitrite (NO_2_) levels of EA.hy926 cells were used to indicate NO production. The nitrite levels were determined following the Griess assay method [47]. The absorbance at 550 nm was measured in a Microplate Biokinetics Reader (Bio-Tek Instruments) and normalized using a standard curve of sodium nitrite levels that was prepared with culture medium.

The PGE_2_ levels in EA.hy926 cells were measured with a competitive enzyme immunoassay (EIA) kit (ADI-900-001, Cayman Chemical, Ann Arbor, MI, USA). The mediator concentrations in the samples were calculated according to standard reference calibration curves.

### 4.6. Intracellular *Interleukin*-*1β* (*IL*-*1β*), *Interleukin*-6 (IL-6), and *Tumor Necrosis Factor-α* (*TNF*-*α*) Level Assessment

At the end of the experimental treatment, the reaction was stopped by removing the medium. The medium removed from each group was used to measure the IL-1β, IL-6, or TNF-α levels. In this study, the IL-1β, IL-6, and TNF-α levels in EA.hy926 cells were measured using Human IL-1 β/IL-1F2 DuoSet ELISA (R&D, DY201-05), Human IL-6 DuoSet ELISA (R&D, DY206-05), and Human TNF-α DuoSet ELISA (R&D, DY210-05) kits (R&D Systems, Inc., Minneapolis, MN, USA) according to the manufacturer’s instructions.

### 4.7. Intracellular BMP-2, Wnt, β-Catenin, p52, and p100 Content Analysis by Immunocytochemistry

At the end of the experimental treatment, the reaction was stopped by removing the medium and washing the cells with PBS. For immunostaining, the cells were fixed with ice-cold 5% paraformaldehyde for 20 min at 4 °C. The cells were incubated with 0.5% Triton X-100 in PBS for 15 min and 1% bovine serum albumin for 30 min. The primary antibodies used for immunostaining were primary antibodies against BMP-2 (1:1000, AF5163, Affinity Biosciences, Cincinnati, OH, USA), Wnt (1:1000, DF2441, Affinity Biosciences), β-catenin (1:1000, BF8016, Affinity Biosciences), p52 (1:1000, AF3375, Affinity Biosciences), or p100 (1:1000, AF3375, Affinity Biosciences) overnight at 4 °C. The cells were washed with PBS and incubated with an Alexa Fluor 488-conjugated goat anti-rabbit IgG (H+L) highly cross-adsorbed secondary antibody (green, 1:1000, A-11034, Thermo Fisher Scientific Inc., Waltham, MA, USA) for 2 h at room temperature in the dark. The cells were counterstained with 1 µg/mL Hoechst 33342 (Blue, Sigma-Aldrich Co.). After Hoechst 33258 (H33258) was added to the cells, they were washed three times with PBS to remove the excess Hoechst stain. Gen5 Software was used to automate image capture with a Cytation™ 1 Cell Imaging Multi-Mode Reader (BioTek. Instruments). The quantitative value of the fluorescence intensity of a specific target protein was calculated as follows: target protein fluorescence intensity/Hoechst 33258 fluorescence intensity. This allowed the normalization of target protein fluorescence intensity to the relative numbers of cells in each group.

### 4.8. Immunoblotting Analysis of Intracellular BMP-2, p52, and p100 Protein Expression

At the end of the experimental treatment, the reaction was stopped by removing the medium and washing the cells with cold PBS. Then, the cells were harvested in 200 μL of lysis buffer (10 mM Tris-HCl, 5 mM EDTA, 0.2 mM phenylmethylsulfonyl fluoride, and 20 μg/mL aprotinin at pH 7.4). The total protein levels in the EA.hy926 cells were measured using the method described by Lowry et al. [47].

Equivalent amounts of cellular protein (approximately 20 μg) from each sample were separated by 10% sodium dodecyl sulfate (SDS) polyacrylamide gel electrophoresis [47] and transferred to polyvinylidene difluoride membranes [47]. The membranes were then incubated with anti-BMP-2 (1:1000, AF5163, Affinity Biosciences), anti-p52 (1:2000, AF3375, Affinity Biosciences), or anti-p100 (1:200, AF3375, Affinity Biosciences) antibodies at 4 °C overnight, followed by incubation with a peroxidase-conjugated secondary antibody. The bands were visualized using an enhanced chemiluminescence detection kit (RPN3243, Amersham Life Science, Buckinghamshire, UK). For densitometric analysis, the blots were treated with enhanced chemiluminescence substrate solutions and exposed using a ChemiDoc XRSþ System (Bio-Rad Laboratories, Hercules, CA, USA).

### 4.9. Statistical Analysis

All the experiments in this study were performed with three or more replicates per group. The data were analyzed using the statistical analysis software SPSS for Windows, version 20.0 (SPSS Inc., Chicago, IL, USA). One-way analysis of variance (ANOVA) and Tukey’s multiple range tests were used to evaluate the significance of differences between mean values. Except for the intracellular calcium assay, ^abcd^ Values are significantly different from those of the other groups. A *p* value less than 0.05 indicated a statistically significant difference.

## Figures and Tables

**Figure 1 plants-11-03265-f001:**
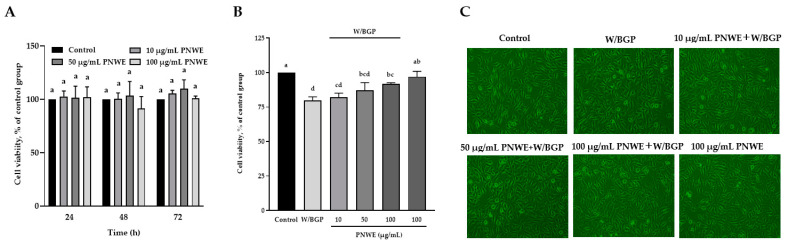
Effect of PNWE on the viability of EA.hy926 cells. EA.hy926 cells (1.0 × 10^5^ cells/30 mm plate) were seeded and cultured overnight. (**A**) EA.hy926 cells were treated with 10, 50, or 100 μg/mL PNWE for 24, 48, and 72 h; (**B**) EA.hy926 cells were treated with 10, 50, or 100 μg/mL PNWE for 48 h followed by stimulation with or without 10 µM warfarin/1.6 mM β-glycerophosphate (W/BPG) for another 24 h; (**C**) morphological examination of EA.hy926 cells after the above treatments. The cells treated with sterilized RO H_2_O alone for 72 h served as the control group, and cells treated with sterilized RO H_2_O alone for 48 h followed by treatment with W/BGP for the last 24 h served as the W/BGP-treated control group. Cells treated with 100 μg/mL PNWE for 72 h served as another control group. The values are presented as the means ± SDs (*n* = 3). ^abcd^ Values not sharing the same letter are significantly different, as shown by Tukey’s test (*p* < 0.05) in the cell viability assay.

**Figure 2 plants-11-03265-f002:**
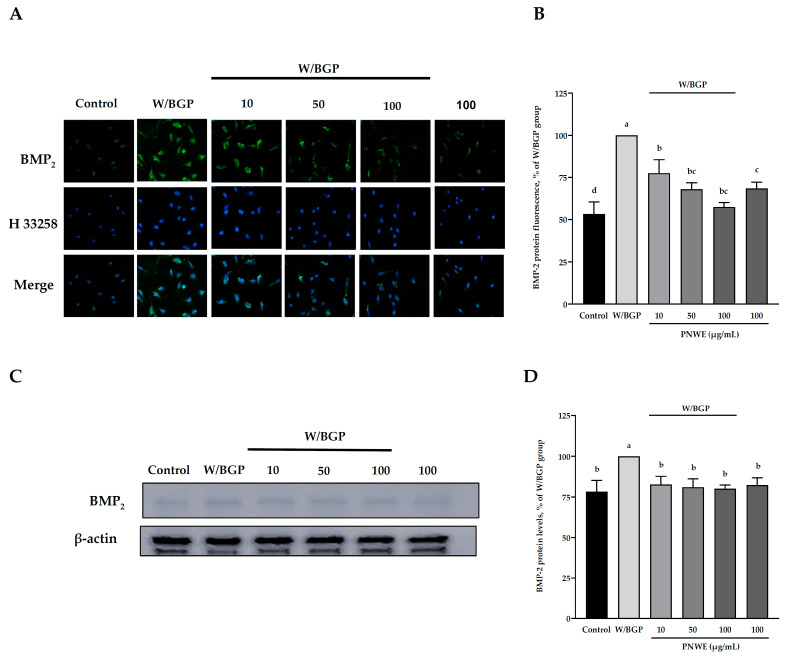
Effect of PNWE on BMP-2 protein expression in EA.hy926 cells. EA.hy926 cells (1.0 × 10^5^ cells/30 mm plate) were seeded and cultured overnight. EA.hy926 cells were treated with 10, 50, or 100 μg/mL PNWE for 48 h followed by stimulation with or without 10 µM warfarin/1.6 mM β-glycerophosphate (W/BPG) for another 24 h. (**A**) Immunocytochemical staining for cytosolic BMP-2 protein (green), nuclei stained with Hoechst 33258 (blue), and a merged image. (**B**) Quantitative analysis of the relative levels of BMP-2 in the figure. (**C**) Immunoblotting assays were performed to determine the expression levels of BMP-2 in EA.hy926 cells. (**D**) Quantitative analysis of the protein levels of BMP-2 in the figure. The protein expression in EA.hy926 cells was quantified using densitometry. The expression levels in the W/BPG-treated group were set to 100%. The values are presented as the means ± SDs (*n* = 3). ^abcd^ Values not sharing the same letter are significantly different, as shown by Tukey’s test (*p* < 0.05).

**Figure 3 plants-11-03265-f003:**
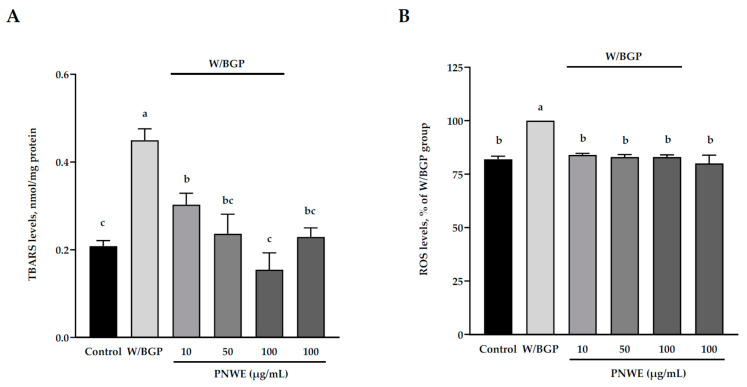
Effect of PNWE on oxidative stress in EA.hy926 cells. EA.hy926 cells (1.0 × 10^5^ cells/30 mm plate) were seeded and cultured overnight. EA.hy926 cells were treated with 10, 50, or 100 μg/mL PNWE for 48 h followed by stimulation with or without 10 µM warfarin/1.6 mM β-glycerophosphate (W/BPG) for another 24 h. (**A**) TBARS levels; (**B**) ROS levels. The expression levels in the W/BPG-treated group were set to 100%. The values are presented as the means ± SDs (*n* = 3). ^abc^ Values not sharing the same letter are significantly different, as shown by Tukey’s test (*p* < 0.05).

**Figure 4 plants-11-03265-f004:**
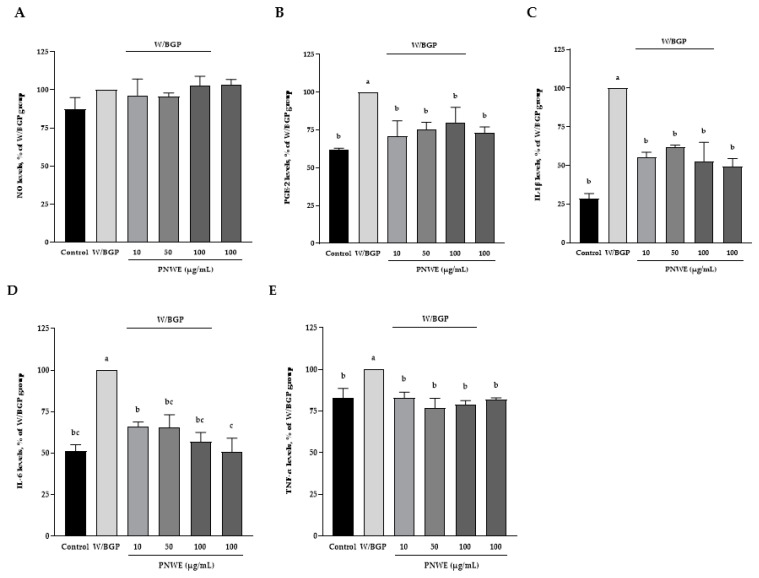
Effect of PNWE on the inflammatory response in EA.hy926 cells. EA.hy926 cells (1.0 × 10^5^ cells/30 mm plate) were seeded and cultured overnight. EA.hy926 cells were treated with 10, 50, or 100 μg/mL PNWE for 48 h followed by stimulation with or without 10 µM warfarin/1.6 mM β-glycerophosphate (W/BPG) for another 24 h. (**A**) NO levels; (**B**) PGE_2_ levels; (**C**) IL-1β levels; (**D**) IL-6 levels; (**E**) TNF-α levels. The expression levels in the W/BPG-treated group were set to 100%. The values are presented as the means ± SDs (*n* = 3). ^abc^ Values not sharing the same letter are significantly different, as shown by Tukey’s test (*p* < 0.05).

**Figure 5 plants-11-03265-f005:**
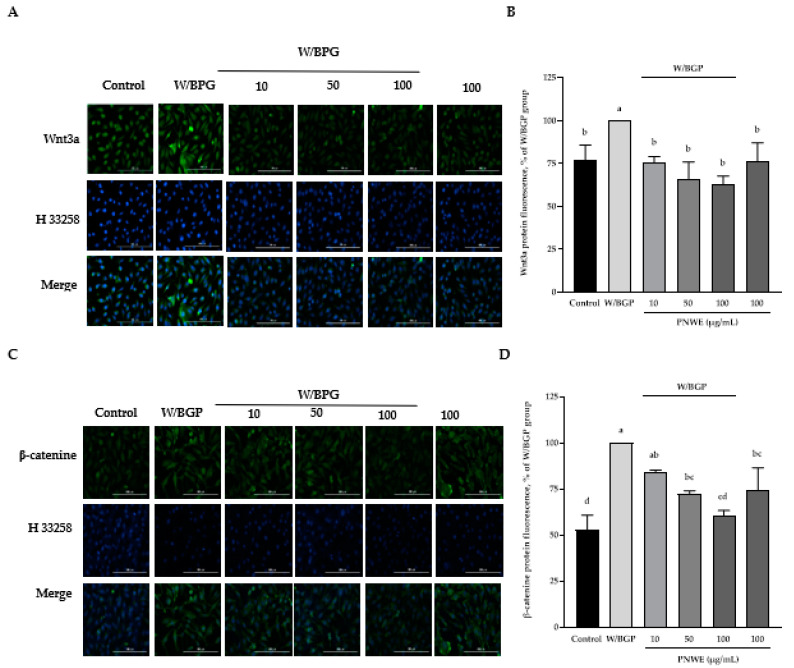
Effect of PNWE on Wnt/β-catenin protein expression in EA.hy926 cells. EA.hy926 cells (1.0 × 10^5^ cells/30 mm plate) were seeded and cultured overnight. EA.hy926 cells were treated with 10, 50, or 100 μg/mL PNWE for 48 h followed by stimulation with or without 10 µM warfarin/1.6 mM β-glycerophosphate (W/BPG) for another 24 h. (**A**) Immunocytochemical staining for cytosolic Wnt3a protein (green), nuclei stained with Hoechst 33258 (blue), and a merged image. (**B**) Quantitative analysis of the relative levels of Wnt3a in the figure. (**C**) Immunocytochemical staining for cytosolic β-catenin protein (green), nuclei stained with Hoechst 33258 (blue), and a merged image. (**D**) Quantitative analysis of the relative levels of β-catenin in the figure. The expression levels in the W/BPG-treated group were set to 100%. The values are presented as the means ± SDs (*n* = 3). ^abcd^ Values not sharing the same letter are significantly different, as shown by Tukey’s test (*p* < 0.05).

**Figure 6 plants-11-03265-f006:**
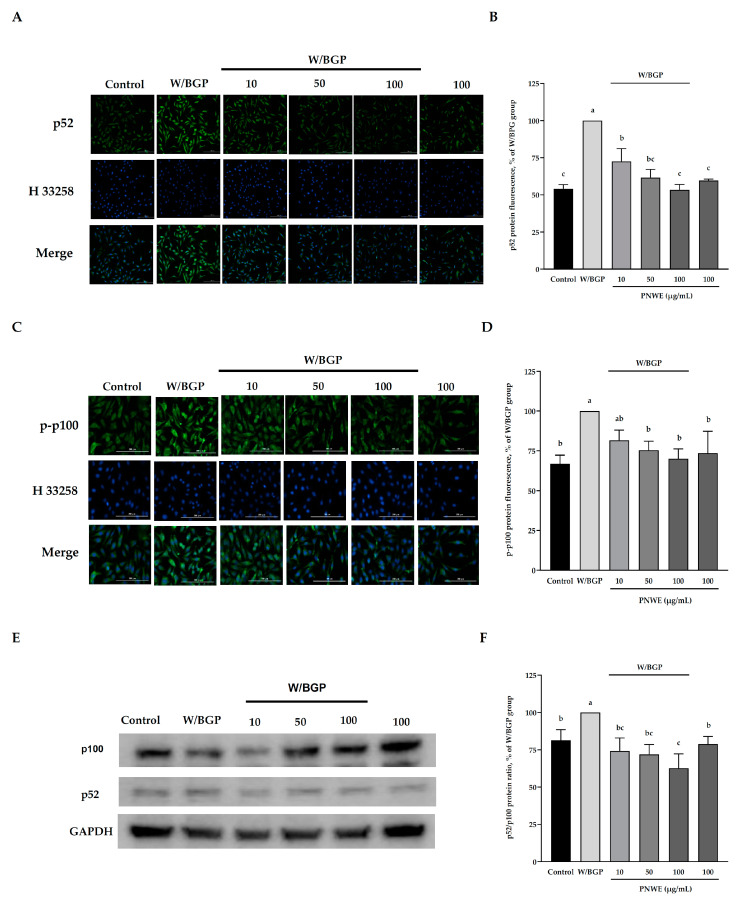
Effect of PNWE on noncanonical NF-κB protein expression in EA.hy926 cells. EA.hy926 cells (1.0 × 10^5^ cells/30 mm plate) were seeded and cultured overnight. EA.hy926 cells were treated with 10, 50, or 100 μg/mL PNWE for 48 h followed by stimulation with or without 10 µM warfarin/1.6 mM β-glycerophosphate (W/BPG) for another 24 h. (**A**) Immunocytochemical staining for cytosolic p52 protein (green), nuclei stained with Hoechst 33258 (blue), and a merged image; (**B**) quantitative analysis of the relative levels of p52 in the figure; (**C**) immunocytochemical staining for cytosolic p-p100 protein (green), nuclei stained with Hoechst 33258 (blue), and a merged image; (**D**) quantitative analysis of the relative levels of p-p100 in the figure; (**E**) immunoblot assays were performed to determine the expression levels of p100 and p52; (**F**) the p52/p100 ratio in EA.hy926 cells. The protein expression in EA.hy926 cells was quantified using densitometry. The expression levels in the W/BPG-treated group were set to 100%. The values are presented as the means ± SDs (*n* = 3). ^abc^ Values not sharing the same letter are significantly different, as shown by Tukey’s test (*p* < 0.05).

**Figure 7 plants-11-03265-f007:**
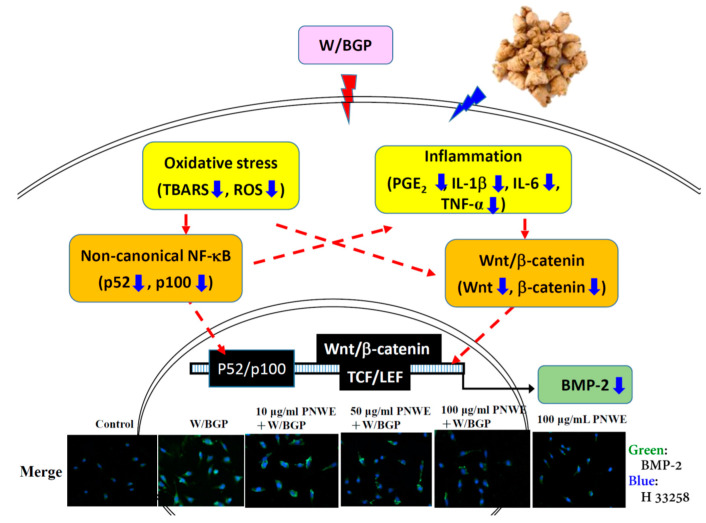
Possible mechanisms by which PNWE inhibits warfarin/β-glycerophosphate-induced BMP-2 expression in EA.hy926 cells.

## Data Availability

The datasets used and/or analyzed during the current study are available from the corresponding author on reasonable request.

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
