# Peer review of "Panax notoginseng* Suppresses Bone Morphogenetic Protein-2 Expression in EA.hy926 Endothelial Cells by Inhibiting the Noncanonical NF-κB and Wnt/β-Catenin Signaling Pathways"

_plants, 2022, doi:10.3390/plants11233265_

Round 1
Reviewer 1 Report
The article is interesting and may interest the magazine's readers. The topic of the article is very topical. The presented research seems to form a logical whole. The research methodology does not raise my objections. I have reservations about the literature cited. It is very outdated and requires enrichment with new Works. There is no chapter explaining the abbreviations used throughout the article. There are a lot of them, and this makes it difficult to read the article. The article summary and the discussion of the results are missing. Without these two chapters, the article cannot be published.
Author Response
Response to Reviewer 1 Comments
Point 1: The article is interesting and may interest the magazine's readers. The topic of the article is very topical. The presented research seems to form a logical whole. The research methodology does not raise my objections. I have reservations about the literature cited. It is very outdated and requires enrichment with new Works. There is no chapter explaining the abbreviations used throughout the article. There are a lot of them, and this makes it difficult to read the article. The article summary and the discussion of the results are missing. Without these two chapters, the article cannot be published.
Response 1:
- We thank the reviewer for the comments. We have revised and upgraded the early references, such as reference nos. 2, 11, 13, 16, 17, 23, and 47.
- We thank the reviewer for the comments. We added an abbreviations list on page 13, lines 495-502.
- The summary is presented as an abstract following the title page. The “Discussion” section is presented on page 9, line 272.

Reviewer 2 Report
The article is interesting and relevant to the special issue, but the poor English makes it much more difficult to read.
Lines 76-77 "To study the potential of PN in reducing VC progression. In this study, PN water extract (PNWE) was prepared by ourselves through ultrasonic extraction, vacuum concentration and lyophilization as experimental materials." - obviously it should be one sentence, not two.
Some bars on the graphs do not contain the statistical error of the mean data. Such accuracy could be true if it were not repeated only in the control as in Figures 1a and 1b. Please, check all data.
Figure 6 should be removed from the Discussion.
The Discussion about the mechanism of development of vascular diseases seems redundant. I would like the authors to discuss what components of the PN extract make such a biological effect possible. In addition, parallels with Panax ginseng are welcome in the Discussion or Introduction.
Author Response
Response to Reviewer 2 Comments
Point 1: The article is interesting and relevant to the special issue, but the poor English makes it much more difficult to read.
Response 1:
We thank the reviewer for the comments. We have had the manuscript re-edited by a professional language editor. The certified documents are in the attached file.
Point 2: Lines 76-77 "To study the potential of PN in reducing VC progression. In this study, PN water extract (PNWE) was prepared by ourselves through ultrasonic extraction, vacuum concentration and lyophilization as experimental materials." - obviously it should be one sentence, not two.
Response 2: We thank the reviewer for the comments. We have revised the text as follows: “To study the potential of PN in reducing VC progression, we prepared PN water extract (PNWE) through ultrasonic extraction, vacuum concentration and lyophilization as experimental material in this study.” Page 2, lines 78-81.
Point 3: Some bars on the graphs do not contain the statistical error of the mean data. Such accuracy could be true if it were not repeated only in the control as in Figures 1a and 1b. Please, check all data.
Response 3:We thank the reviewer for the comments. Some figures are presented as the percentage of the control or the inducer (W/BGP) group. The standard deviation value of the control or the inducer (W/BGP) group was zero. Therefore, the statistical error bars in the graphs cannot be seen.
We also rechecked all data and figures. We revised Figure 1A, even though it was not significantly different between each group.
Point 4: Figure 6 should be removed from the Discussion.
Response 4:Thank you for the comment. We relocated Figure 6 to page 8 so that it is not in the “Discussion” section.
Point 5: The Discussion about the mechanism of development of vascular diseases seems redundant. I would like the authors to discuss what components of the PN extract make such a biological effect possible. In addition, parallels with Panax ginseng are welcome in the Discussion or Introduction.
Response 5: Thank you for the comment. We added some discussion of the effects of PN and Panax ginseng and their components on regulating the inflammatory response, oxidative stress, and vascular calcification. The potential and possible mechanism of PNWE in reducing vascular calcification is described (page 9, line 299 to page 10, line 307).

Reviewer 3 Report
Dear author(s), I would like to appreciate your detailed research study particularly detailed discussion with appropriate research design and references. Very impressive work. I have only one question. Did you check the presence of ginsenoside Rb1 and Rg1 in your water extract? If it is present in your extract then provide the concentartions of these compounds.
Author Response
Reviewer 3
Dear author(s), I would like to appreciate your detailed research study particularly detailed discussion with appropriate research design and references. Very impressive work. I have only one question. Did you check the presence of ginsenoside Rb1 and Rg1 in your water extract? If it is present in your extract then provide the concentartions of these compounds.
Thanks for the comment.
In our study, there are analyzed the contents of notoginsenoside R1 and ginsenoside Rg1 in PNWE, that as indicators of different repeat extraction qualities. We add these data in "4.1. Preparation of PN water extract (PNWE)" (page 12, lines 378-380).
Round 2
Reviewer 1 Report
The article is correctly written. It includes a complete set of research and analyzes. The research methodology is correct. The article may be accepted for publication.
Author Response
Reviewer 1
The article is correctly written.It includes a complete set of research and analyzes. The research methodology is correct. The article may be accepted for publication.
Thanks for your affirmative.